environmental science, theoretical biology, ecology

cumulative effects, synergism, antagonism, meta-analysis, cumulative impacts

**Author for correspondence:**
Cody J. Dey
e-mail: cody.dey@dfo-mpo.gc.ca

# The consequences of null model selection for predicting mortality from multiple stressors

Cody J. Dey and Marten A. Koops

Great Lakes Laboratory for Fisheries and Aquatic Sciences, Fisheries and Oceans Canada, 867 Lakeshore Road, Burlington, ON, Canada, L7S 1A1

CJD, 0000-0003-4947-8972; MAK, 0000-0002-3676-7946

Many ecological systems are now exposed to multiple stressors, and ecosystem management increasingly requires consideration of the joint effects of multiple stressors on focal populations, communities and ecosystems. In the absence of empirical data, ecosystem managers could use null models based on the combination of independently acting stressors to estimate the joint effects of multiple stressors. Here, we used a simulation study and a meta-analysis to explore the consequences of null model selection for the prediction of mortality resulting from exposure to two stressors. Comparing five existing null models, we show that some null models systematically predict lower mortality rates than others, with predicted mortality rates up to 67.5% higher or 50% lower than the commonly used Simple Addition model. However, the null model predicting the highest mortality rate differed across parameter sets, and therefore there is no general 'precautionary null model' for multiple stressors. Using a multi-model framework, we re-analysed data from two earlier meta-analyses and found that 54% of the observed joint effects fell within the range of predictions from the suite of null models. Furthermore, we found that most null models systematically underestimated the observed joint effects, with only the Stressor Addition model showing a bias for overestimation. Finally, we found that the intensity of individual stressors was the strongest predictor of the magnitude of the joint effect across all null models. As a result, studies characterizing the effects of individuals stressors are still required for accurate prediction of mortality resulting from multiple stressors.

## 1. Introduction

The anthropocene has been characterized by widespread changes to the natural world. Most ecosystems are now exposed to multiple stressors, which can have cumulative effects that can drive profound changes in ecosystem function and biodiversity [1,2]. As such, the management of multiple stressors and their cumulative effects is an issue from local (e.g. [3]) to global scales (e.g. Convention on Biological Diversity Aichi Target 10 [4]). Addressing multiple stressors is inherently complicated due to the complexity of natural systems, the variety of stressors acting on ecosystems, and the variance in individual, population and community responses to those stressors [5]. Yet, despite these challenges, many decision-makers are legislatively required to consider the cumulative effects of multiple stressors, and therefore improving the scientific understanding of multiple stressors and their interactions is an important focus for researchers [6].

An important part of understanding the cumulative effects of multiple stressors is understanding whether stressors will have some mechanistic (e.g. physical or physiological) interaction that would impact populations or ecosystems more (i.e. synergistic) or less (i.e. antagonistic) severely relative to the sum

of the independent effects of each stressor (e.g. [7–9]; see [10] for a review). The implication is that synergistic stressor combinations represent 'ecological surprises' that can complicate environmental management because of their potential to accelerate biodiversity loss and impair the functioning of ecosystems [2,11,12]. Conversely, unexpected antagonistic combinations may help managers prevent severe adverse losses, such that understanding which effect to expect is critical.

However, the determination of synergistic or antagonistic stressor combinations can only be made by comparing the realized impact of multiple stressors to that predicted by a particular null model [13,14]. Importantly, there are a range of null models predicting the joint effect of independently acting stressors [15–17] which vary in their predictions for a given set of parameter values (i.e. given individual stress effects; see figure 1). As such, an observed effect deemed to be synergistic in reference to one null model may be consistent with (or even antagonistic to) a second null model. Because different studies may use different null models, it is challenging to derive general trends about the probability that multiple stressors produce synergistic impacts. Furthermore, because most studies only consider a single null model (e.g. [18–22]), the impression that the joint effects of multiple stressors are often synergistic or antagonistic could be a result of an incomplete consideration of the range of potential null models. Finally, among-study differences in analytic approaches complicate the interpretation of the predictive ability of various null models (see box 1), and the frequency of synergistic and antagonistic stressor effects.

In ecological studies, the Simple Addition model is often chosen as the null model against which joint stress effects are tested [14,19,26]. This model assumes that the joint effects of multiple stressors will be equal to the sum of the effects of the individual stressors. While Simple Addition represents an intuitive null model (and is invoked by certain statistical tests; see [16]); it has features that make it undesirable as a default null model. In particular, when mortality is used as the endpoint of interest, Simple Addition assumes there is a strong negative correlation in the distribution of sensitivity to each stressor (i.e. individuals that are sensitive to stressor A are insensitive to stressor B, and vice versa). However, this assumption is not supported by any empirical research to our knowledge, and is opposed by studies showing across-context repeatability in the physiological stress response in many species [27]. Furthermore, a recent study by Thompson et al. [17] demonstrated that when the Simple Addition model is applied to community-level properties, it does not properly aggregate the responses of individual species to produce a linear expectation at the community level. Finally, predictions of mortality for multiple stressors under Simple Addition can exceed 100% (e.g. if stressor A causes 55% mortality alone, and stressor B causes 50% mortality alone) and as such Simple Addition produces impossible mortality predictions in some circumstances [10] (see also [17]).

Other null models from the ecological and ecotoxicological literature are also based on the independent action of each stressor but with different assumptions. For example, the Multiplicative model is based on the assumption that stressors are additive in terms of their probabilistic sum (table 1), which implies that the population's sensitivity to mortality from different stressors is uncorrelated [16].

Alternatively, the Dominance model assumes that the stressor with the largest effect determines the outcome of joint stress effects (table 1), implying a strong positive correlation in the sensitivity to different stressors. The Concentration Addition model is widely used in the field of (eco)toxicology, and assumes that the intensity of stressors (or the concentration of chemicals) is exchangeable, if they are scaled by their potency [31]. This model requires understanding the relationship between stressor intensity and the endpoint of interest (which we term the stressor-effect relationship, box 1) through either empirical means, or by assuming its shape. Finally, the Stressor Addition model assumes that individuals have a set tolerance towards all types of stress (their general stress capacity) and that all stressors can be translated into general stress levels, which act as a common currency to combine the effects of multiple stressors [32]. The set of models considered in this study are not a complete set of all possible null models for the joint effect of multiple stressors, and we expect that additional null models will be developed in the future. However, our study represents an attempt to move towards a multi-model framework in a field that has traditionally focused on the Simple Addition model as a default expectation for the comparison of multiple stressor effects.

In this study, we quantify the mortality predictions made by different null models for populations exposed to two stressors. We focus on mortality because it has a strong direct relationship with population dynamics, and is, therefore, often a focus for ecosystem managers. However, four out of the five null models considered (all except the Stressor Addition model) have direct applications for continuous endpoints such as growth or reproduction (see [16] for examples), and as such, our study may provide general insights for the consideration of multiple stressors. In our study we use two complementary approaches, as follows. (i) We examine the range of predictions made by five null models under different stressor-effect relationships and stressor intensities using simulated data. Such an examination is critical for understanding the conditions under which null model selection is consequential (i.e. null models diverge) or inconsequential (i.e. null models converge). In conducting these analyses, we compare the predictions of the Multiplicative, Dominance, Concentration Addition and Stressor Addition null models to the Simple Addition model, because this is the most commonly applied null model in ecological studies of multiple stressors. Additionally, we evaluate whether any of the null models considered in this study could be considered as a precautionary null model for policy and management decisions that rely upon the precautionary principle [33], which we define as a null model that consistently predicts the highest level of impacts (i.e. highest mortality rate). (ii) We re-analyse data presented in previous meta-analyses of multiple stressor effects [14,19]. In doing so, we consider if the observed joint effects are consistent with any of the five null models we consider in the current study, which differs from the common approach of only comparing the observed joint effects with those predicted by the Simple Addition model (e.g. [19,22,26]). As such, this analysis allows us to quantify what proportion of stressor combinations could be explained by the combined effects of independently acting stressors (i.e. those within the range of the null models), and identify stressor combinations that are synergistic or antagonistic with respect to all null models.

**Box 1.** A conceptual framework for predicting the joint effect of multiple stressors

*Functional definitions used in this study*

*Stressor*: changes in environmental parameters beyond their usual range (often directly or indirectly caused by human activity) that ultimately results in a negative biological response.

*Stressor intensity*: measure of the strength of a stressor (e.g. the concentration of a toxic chemical or the increase in temperature of a waterbody). In our simulations, stressor intensities are normalized to the unit scale (0,1), where stressor intensity values of 0 indicate the absence of the stressor and stressor intensity values of 1 relate to the maximum effect of the stressor (e.g. 100% mortality). $SI_A$ and $SI_B$ in table 1.

*Stressor-effect relationship*: the functional relationship between the stressor intensity (e.g. the concentration of a pollutant) and the population response (here, the population mortality rate). $f_A$ and $f_B$ in table 1.

*Individual stress effect*: the population mortality rate caused by exposure to a single stressor, at a given stressor intensity. $f_A(SI_A)$ and $f_B(SI_B)$ in table 1.

*Joint stressor effect* (*joint effect*): the mortality rate experienced by a population exposed to two (or more) stressors, at given stressor intensities. $f_{AB}(SI_A, SI_B)$ in table 1.

*Null models for multiple stressors*: models predicting the joint effect of multiple stressors based on the independent action of each stressor (i.e. stressors do not have any explicit mechanistic interactions).

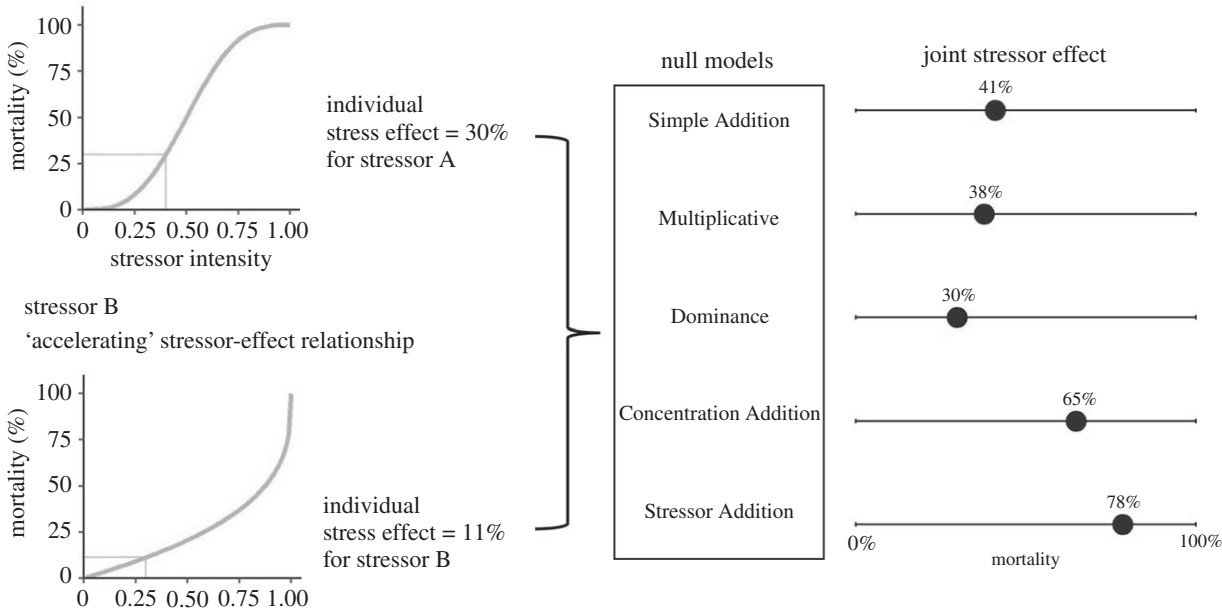

**Figure 1.** Conceptual framework showing the relationship between stressor intensities, stressor-effect relationships, individual stress effects, null models and the predicted joint stressor effects under those null models, for two stressors acting on a population. Stressor intensities determine the individual stress effects, as mediated by the stressor-effect relationships. The manner in which individual stress effects combine to create a joint effect is described by the five null models considered in this study. In this example, stressor intensities of 0.40 (for stressor A) and 0.30 (for stressor B) result in individual stress effects of 30% and 11% mortality, respectively. Null models of multiple stressors predict that a population exposed to both of these stressors will experience mortality between 30% (Dominance model) and 78% (Stressor Addition model).

**Table 1.** Null models for the joint effects of two stressors (see box 1 for a description of each variable in the model equations, and see [16] for a detailed description of each model).

| model | also called | model equation | reference |
|---|---|---|---|
| Simple Addition | Additive, Linear Addition | $f_{AB}(SI_A, SI_B) = f_A(SI_A) + f_B(SI_B)$ | [28] |
| Multiplicative[a] | Effect Addition, Risk model | $f_{AB}(SI_A, SI_B) = f_A(SI_A) + f_B(SI_B) - f_A(SI_A) f_B(SI_B)$ | [29] |
| Dominance | Comparative Effects | $f_{AB}(SI_A, SI_B) = \max(f_A(SI_A), f_B(SI_B))$ | [30] |
| Concentration Addition[b] | — | $f_{AB}(SI_A, SI_B) = f_A(SI_A + \gamma SI_B)$ | [31] |
| Stressor Addition[c] | — | $f_{AB}(SI_A, SI_B) = F_{strcap}(F_{strcap}^{-1}(f_A(SI_A)) + F_{strcap}^{-1}(f_B(SI_B)))$ | [32] |

[a]In the Multiplicative model, individual and joint stress effects must be specified as proportions [13].

[b]With $\gamma = SI_{Ax}/SI_{Bx}$ where $f_A(SI_{Ax}) = f_B(SI_{Bx}) =$ half the effect size limit.

[c]$F_{strcap}$ is the cumulative density function of the population's stress capacity, while $F_{strcap}^{-1}$ is the quantile function for the population's stress capacity.

*Null models in a regression context*

Many studies have used null hypothesis significance tests based on (generalized) linear regressions to test whether the observed effects of multiple stressors adhere to the predictions of various models. Indeed, an alternative idea of a null model for multiple stressor is a simplified version of a statistical model (e.g. an intercept-only statistical model, or a model without statistical interaction effects). This approach differs from that used in the current paper, and the relative popularity of regression approaches versus the predictive approach used herein differs between different subfields [23]. One complication associated with the regression approach is that the use of link functions (or data transformations) to meet the assumptions of the statistical models alters the form of the null model being tested. For example, the common process of log-transforming data prior to conducting an ANOVA imposes the Multiplicative null model rather than the Simple Addition model [16,24], a distinction that is often not appreciated and that can lead to incorrect inferences related to how stressors combine [25]. Many regression approaches also impose a linear (or linear on the link scale) relationship between the intensity of stressors and the biological response, however, stressor-effect relationships follow many possible shapes. Notwithstanding these issues, regression approaches are often more appropriate in observational studies where the intensities of individual stressors can be measured, but the effects of individual stressors (i.e. in isolation) are often unknown.

## 2. Methods

### (a) Null model simulations

To examine the range of joint effects predicted by different null models, we simulated stressor intensities ($SI_A$ and $SI_B$) and stressor-effect relationships ($f_A$ and $f_B$) for two stressors and calculated the predicted joint stressor effect under the five null models presented in table 1. A single simulation was conducted for each unique set of parameter values including stressor intensities from 0 to 1 (at intervals of 0.01) and stressor-effect relationships that were one of five shapes (linear, accelerating, diminishing, steep middle, steep extremes—see electronic supplementary material, figure S1 for more details). The distribution of stress capacities ($F_{strcap}$) used in the Stressor Addition model was parameterized using a symmetric beta distribution with $\alpha = \beta = 3.2$ as per Liess *et al*. [32]. This analysis, therefore, included 250 000 unique parameter sets which collectively represent a broad suite of possible conditions across which resource managers may need to consider the joint effects of two stressors including individual stressors that cause anywhere between 0% and 100% mortality, and a suite of common stressor-effect relationships.

We note that an analytical approach could be used to produce some of the insights from these simulations. For example, comparing the equations in table 1 demonstrates that the Dominance model will always predict a joint effect equal to or less than the prediction from the Simple Addition and Multiplicative null models (similarly, the Dominance model always predicts a lower or equal joint effect to the Multiplicative model). However, an analytical consideration of all null models in this study is complicated by variation in stressor-effect relationships and the distribution of stress capacities, and for these reasons, we selected a simulation approach.

We quantified model divergence as the absolute difference in percentage mortality between the null model predicting the highest and lowest mortality rates, for a given parameter set. Similarly, the model deviation was measured as the standard deviation in mortality rates across null models. High values of model deviation and divergence, therefore, suggest that null model selection is consequential, while low values suggest that all null models make similar predictions. In all cases, mortality predictions greater than 100% (e.g. from the Simple Addition model), were truncated to 100% prior to calculations and plotting.

### (b) Meta-analysis

To explore how empirical data on the joint effects of multiple stressors compare to the joint effects predicted by null models, we reanalysed the data compiled by Crain *et al*. [19] and Darling & Côté [14]. These authors conducted meta-analyses on the effects of two stressors acting in combination, and required the contributing experiments to be conducted in a full factorial design (i.e. to

have measured the response in (i) a control group, (ii) a group exposed to just stressor A, (iii) a group exposed to just stressor B and (iv) a group exposed to stressors A and B in combination). We subsetted the experiments included in the Crain *et al*. [19] and Darling & Côté [14] datasets that measured the effects of two stressors on survival or mortality in animal or plant populations ($n = 170$). We further filtered the data by excluding experiments in which exposure to either of the 'stressors' resulted in an increase in survival ($n = 49$), as these effects were not consistent with our definition of a stressor (see box 1). For the remaining data ($n = 121$ experiments), we calculated the individual stress effects (i.e. $f_A(SI_A)$ and $f_B(SI_B)$) and joint stress effects (i.e. $f_{AB}(SI_A, SI_B)$) for each experiment. To do so we converted all values to per cent mortality (e.g. % mortality = 100% − % survival), and then calculated the difference in mortality between the treatment and control groups as the stress effects (e.g. $f_A(SI_A)$ = mortality in the group exposed to just stressor A − mortality in the control group). For each experiment, we then calculated the predicted joint effects under each of the five null models listed in table 1. For the Concentration Addition model, we calculated the predicted joint effects under all combinations of the five stressor-effect relationships included in electronic supplementary material, figure S1, while scaling the stressor-effect relationships to the scope of possible mortality values (e.g. if the control group showed 10% mortality, the maximum stress effect for each stressor-effect relationship was reduced to 90%, while maintaining the shapes in electronic supplementary material, figure S1). For the Stressor Addition model, we used a symmetric beta distribution ($p = q = 3.2$) for the distribution of general stress capacity, as above.

For each experiment, we then determined if the observed joint effect fell within or outside the range of joint effects predicted by the five null models. To understand which model most accurately predicted the observed joint stressor effects in our dataset, we calculated the bias and precision of each of the null models relative to the observed joint effects. We calculated bias as the mean difference in the model predictions from the observed joint effects, and the precision as the standard deviation in the difference in the model predictions from the observed joint effects.

### (c) Software

Simulations and data analysis were conducted in R v. 3.6.1 [34] using the *tidyverse* suite of packages [35]. We also used the *distr* package [36] for simulating beta distributions with different characteristics.

## 3. Results

### (a) Null model simulations

Across parameter sets (i.e. given values of stressor intensities and stressor-effect relationships), null models showed a

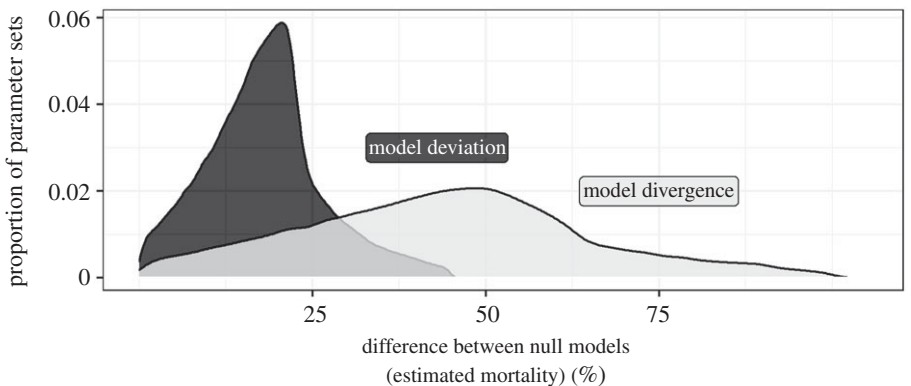

**Figure 2.** The distribution of model deviation (standard deviation of the predicted joint effects) and model divergence (the difference between the maximum and minimum predicted joint effects) across five null models over 250 000 parameter sets. Low values of deviation and divergence are produced when all null models make similar predictions for the joint effect of two stressors, while high values indicate that null models produced different predictions.

**Table 2.** Strength of model input variables as drivers of the amount of predicted mortality across five null models. Each variable listed below was included in a univariate binomial family generalized linear model and the resultant model likelihood compared with likelihood from an intercept only model, in the manner outlined by McFadden [37]. Larger pseudo $R^2$ values indicate that the variable has a stronger impact on the amount of predicted mortality.

| variable | pseudo $R^2$ |
| --- | --- |
| intensity of stressor A | 0.139 |
| intensity of stressor B | 0.137 |
| stressor-effect relationship for stressor A | 0.047 |
| stressor-effect relationship for stressor B | 0.054 |
| null model | 0.044 |

median divergence of 44.0% mortality and a median deviation of 18.0% mortality (figure 2). In general, model divergence and deviation increased as stressor intensities increased, and were higher when the stressor-effect relationships followed a 'diminishing or steep extremes' shape (electronic supplementary material, table S1). In comparison to the commonly used Simple Addition model, other null models predicted population mortality rates up to 67.5% higher or up to 50% lower (electronic supplementary material, figure S3). As expected, Dominance and Multiplicative null models never produced mortality estimates greater than the Simple Addition model. In addition, the Stressor Addition model never produced mortality estimates less than the Simple Addition model. As a result of interactions between the shapes of the two stressor-effect relationships, the Concentration Addition model produced highly variable mortality estimates (electronic supplementary material, figure S3).

These results illustrate the challenge of predicting the joint effects of multiple stressors. Even in simple two stressor systems different stressor-intensities, stressor-effect relationships and null models cause large amounts of variation in the predicted joint effect. To attempt to understand what factors have the most impact on the variance in predicted mortality across models, we used pseudo $R^2$ values to quantify the variance in mortality explained by the different input variables [37]. We found that the stressor intensities were the strongest drivers of variance in predicted mortality (table 2).

Additionally, the two stressor-effect relationships, and the null model used, were about equal contributors to the variation in predicted mortality. However, even when individual stress effects are known (i.e. $f_A(SI_A)$ and $f_B(SI_B)$), variance in the null model used can lead to substantial variance in predicted joint effects (electronic supplementary material, figure S4).

We used our simulated dataset to investigate whether any of the five null models considered herein could be used as a precautionary null model for resource management decisions requiring the prediction of the joint effect of multiple stressors. If a single null model routinely makes the highest mortality predictions, resource managers could use such a model as an upper (i.e. precautionary) prediction for the amount of mortality likely to occur from a given combination of stressors. No single null model in our simulation predicted the highest level of mortality across all parameter sets, and therefore none of the null models considered in this study can be considered a general precautionary null model. The Stressor Addition model was the null model most likely to produce the highest mortality estimate (figure 3a); however, there was a wide area of parameter space over which no null model consistently produced precautionary esimates (figure 3b). These results suggest that the shape of the stressor-effect relationships often determine which null model produces the highest mortality estimates.

## (b) Meta-analysis

In total, 121 experiments included in Crain et al.'s [19] and Darling & Côté's [14] meta-analyses met the criteria for inclusion in our re-analysis. Of these 121 experiments, we found that 54% of the observed joint effects fell within the range of null model predictions (figure 4a), despite only 29% of the observed joint effects being within ±5% mortality of the prediction from the Simple Addition model. In addition, we found that 24% of the observed joint effects exceeded the predictions of all null models and 22% were lower than the predictions of all null models.

On average, most null models predicted less mortality than observed (figure 4b), with biases of −8.0%, −12.9%, −9.9% and −6.9% for the Simple Addition, Dominance, Multiplicative and Concentration Addition models, respectively. The stressor addition model was the only null model to predict higher mortality than observed, on average (bias = +5.7%, precision = 36.3%), and also had the lowest

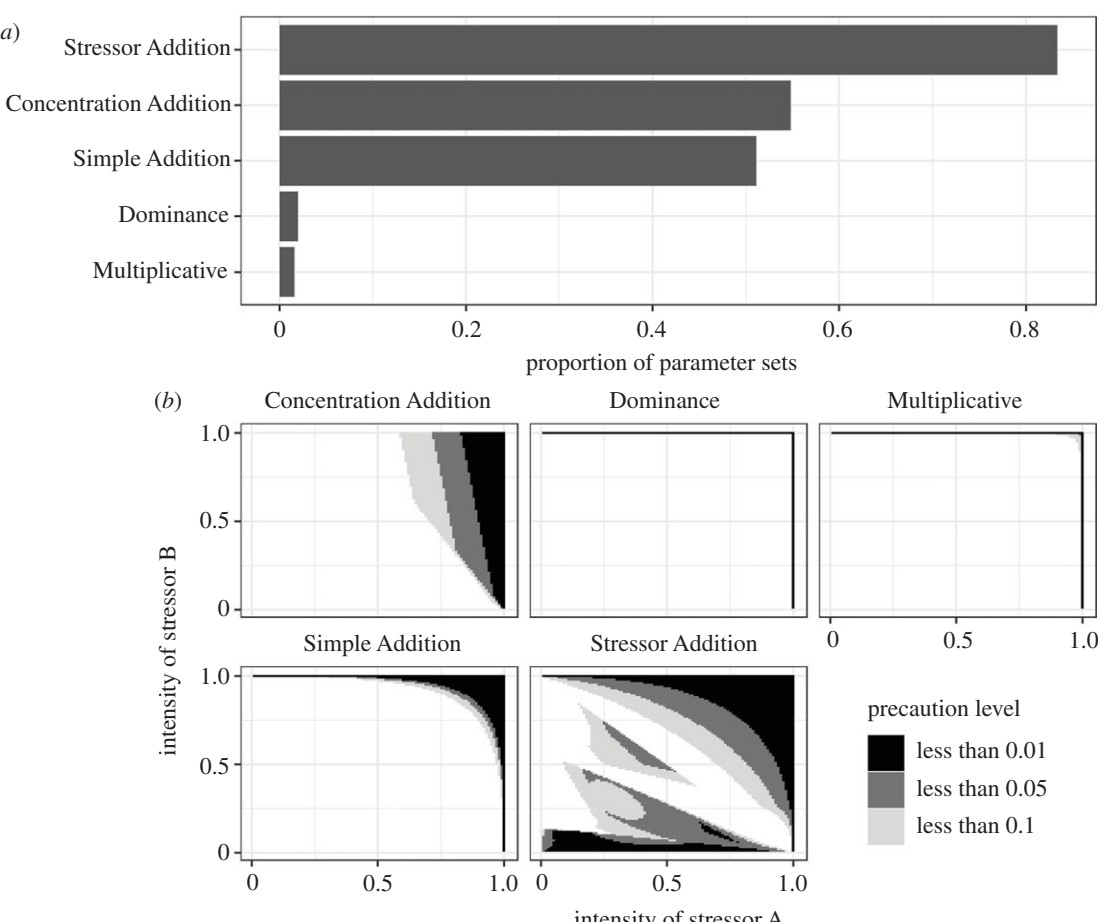

**Figure 3.** Precautionary null models. Here, we show the number of parameter sets in which a particular null model produced predicted mortality values equal to the highest value predicted across all null models (*a*). In addition, we show the parameter space over which a model could be considered precautionary, at different precaution levels (*b*). For example, a model was considered precautionary at the less than 0.05 level if, for all simulation conditions, the model predicts joint effects to be no less than 0.05 (5%) below the maximum predicted joint effect across all null models. The paucity of precautionary null models for low levels of stressor intensities demonstrates that for these combinations of stressor intensities, the shape of the stressor-effect relationships determines which null model predicts the highest mortality rate. As such, no single null model (from the five considered here) can be considered as a precautionary model across all combinations of stressor intensities.

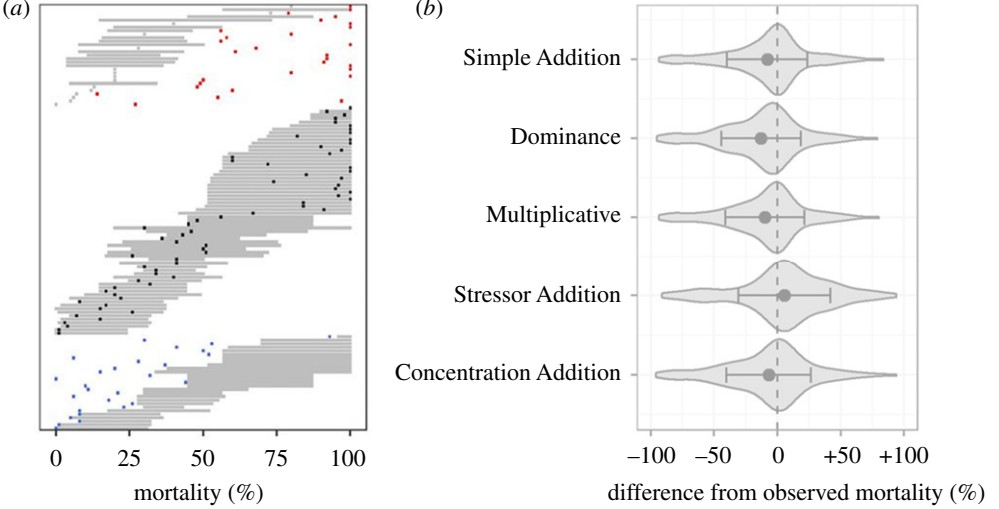

**Figure 4.** Null model predictions for the joint effects of two stressors relative to the effects observed in 121 factorial experiments (*a*). Each row represents a single experiment and shows the range of predicted mortality under the five null models considered in this study (horizontal light grey bars) in comparison to the observed mortality (shown with a coloured square with blue, black or red coloration indicating the observed mortality was less than, within or greater than the range of null model predictions). In (*b*), we show the accuracy of each null model in predicting the observed mortality resulting from two stressors. Grey polygons show the distribution of the difference between the model predictions and the observed mortality, across all 121 experiments, with negative values indicating the model predicted less mortality than observed. Black dots indicate the average difference between the model predictions and the observed mortality (bias), and error bars illustrate the model's precision (standard deviation of the difference).

Proc. R. Soc. B **288**: 20203126

absolute bias. However, the precision of the Stressor Addition model (±36.3%) was lower than all other models (±31.9%, ±31.4%, ±31.3%, ±33.4% for the Simple Addition, Dominance, Multiplicative and Concentration Addition models, respectively), and as such, no clear model can be considered as the most accurate predictor for this dataset.

## 4. Discussion

Null models provide a comparator for observed patterns in nature and can help to challenge scientists' commonsense judgements about how unusual patterns may have been produced [38]. The suite of five null models considered in this study describe five ways (table 1) in which stressors could combine to produce joint effects without any mechanistic interaction between the individual stressors. While some stressors do interact through physical (e.g. the photomodification of polycyclic aromatic hydrocarbons by ultraviolet radiation [39]) or physiological (e.g. competition between toxic metals for cation uptake sites in fish gills [40]) means our simulation study demonstrates that direct interactions between stressors are not theoretically required to produce a range of joint effects for two stressors acting in combination (see also [16]). In our simulation study, we show null model mortality estimates differ by an average of 44% between the models with the most and least severe predictions (figure 2), and that null model predicted joint effects can differ substantially from the commonly used Simple Addition model.

Of the set of null models considered in this study, the Dominance model always produced the lowest mortality predictions. However, no null model systematically produces the highest mortality predictions and therefore, there does not seem to be a general precautionary null model for multiple stressors. As a result, precautionary approaches to estimating mortality from multiple stressors will require engaging with a variety of null models, including grappling with the complexity of the inputs for those null models (e.g. the stressor-effect relationships in the Concentration Addition model and the shape of the general stress capacity in the Stressor Addition Model). Nonetheless, we also found that the intensity of individual stressors remains the strongest predictor of the magnitude of mortality across all null models (table 2). Given that stressor intensities should be easier to quantify or estimate than other model inputs (e.g. stressor-effect relationships), this finding suggests that reasonable model predictions can be produced even in many data deficient scenarios.

In re-analysing data from previous meta-analyses, we found that the majority (54%) of the observed joint effects of multiple stressors on mortality rates fell within the interval of predictions from the set of null models (figure 4a), but that most null models were biased towards underestimating the observed effects (figure 4b). Of the null models considered here, the Stressor Addition model had the lowest absolute bias, predicting 5.7% more mortality than observed on average. However, this model also under-predicted mortality in some experiments by up to 91.3%, and over-predicted mortality by up to 93.8% in other experiments (figure 4b). As a result, we caution that prediction of mortality from multiple stressors that are based on single null models are often likely to be incorrect, sometimes dramatically so.

Our approach of comparing experimental results to a suite of null models contrasted with that of previous meta-analyses (e.g. [18,20–22]) and many empirical studies (e.g. [41–43]) which have compared observed joint effects to a single null model (typically the Simple Addition model). We suggest that a multi-model approach, in which the range of predictions made by a set of ecologically reasonable null models is considered, may provide a more useful framework for ecosystem managers who are tasked with estimating mortality from multiple stressors. While this interval will often be wide (e.g. figures 2 and 4a), it is more likely to include the actual outcome of a given combination of stressors than focusing on a single null model for prediction. Furthermore, previous guidance on developing a reasonable set of hypotheses in statistical analyses (e.g. [44]) could be used to eliminate some null models *a priori*, thereby narrowing the scope of the interval.

These results highlight several additional challenges associated with the study of multiple stressors. First, for many parameter sets, null models (which are based on the combination of independently acting stressors) predict a wide range of possible mortality estimates, and therefore any observation within that range could be explained by the independent action of the stressors. Only when the observed joint effect lies outside the range of predictions produced by all null models, might we infer that the stressors have some mechanistic interaction that has led to an unexpected joint effect. However, it is also possible that stressors do have a direct mechanistic interaction and still produce joint effects within the range of predictions of the null models. As a result, we suggest that understanding whether two stressors have a mechanistic interaction is not sufficient to allow for robust prediction of the joint effects of two stressors, and that much more work on mechanistic interactions between different stressors is required before this information is useful for resource management. Instead, we suggest that a focus on the phenomenology of multiple stressor effects (which ignores whether stressors have a direct interaction) is the more fruitful approach to predicting multiple stressor effects at the current time, despite phenomenological approaches being generally thought of as less predictive frameworks [45]. In this argument, we differ from Schäfer & Piggott [16] who suggest that appropriate null models can be selected based on knowledge of the mechanism of action of the individual stressors involved. While such an approach is a promising step to reducing uncertainty surrounding the joint effects of multiple stressors, we suggest that much more empirical support is needed to demonstrate that such a framework offers any predictive capability.

A related issue surrounds the terms 'synergistic' and 'antagonistic' stressor combinations, which are being used with increasing frequency [10] but have inconsistent definitions among studies. Under their original definitions (*sensu* [13]), an observed joint effect may be synergistic with regard to one null model but antagonistic to another null model, because the determination of synergistic and antagonistic stressor combinations was made in reference to a stated null model. However, many authors have used the terms to define stressor combinations in exclusive reference to the Simple Addition model, and have labelled stressor combinations themselves as 'synergistic' or 'antagonistic' when they result in joint effects that are more or less severe than the predictions of that null model. While this latter definition would provide more consistency and precision for the term, it leaves the question of how to differentiate joint effects that are consistent with another null model (e.g. the Multiplicative

model), versus those that fall outside of the range of all null models. Given the general paucity of stressor combinations that adhere to the predictions of the Simple Addition model, it seems prudent to develop terminology to describe stressor combinations that are consistent with other null models. Indeed, as Orr *et al*. [23] discuss in their recent review, ambiguous and inconsistent use of terms is a general issue in the study of multiple stressors that hinders knowledge transfer among disciplines and therefore warrants further attention.

Additionally, our study shows that for some combinations of parameters, all null models produce similar mortality predictions (e.g. when both stressor intensities were low, or both were high; figure 3; electronic supplementary material, figure S4). As a result, stressors that appear to combine in an additive manner (i.e. the observed mortality is *not significantly different* than that predicted by the Simple Addition model) may actually be following a different null model. The practical implication of this finding is that even when the joint effect of a set of stressors has been measured in a given system, we cannot extrapolate the joint effect of those same stressors (in the same system) if the stressor intensities change. For example, Moreno-Marín *et al*. [43] found that a 10°C increase in temperature, 137 µmol photons $s^{-1}$ $m^{-2}$ decrease in light and a 25 µM increase in nitrogen produced joint effects on eelgrass (*Zostera marina*) that were not significantly different from the predictions of the Simple Addition model. However, we cannot infer that these three stressors will combine to produce joint effects that are consistent with the Simple Addition model at other stressor intensities. An improved approach to understanding multiple stressors would involve quantifying which null model best predicts the observed joint effects (a model selection approach) rather than testing whether the observed joint effects differ from a specific null model (a null hypothesis significance testing approach), especially if joint stressor effects were measured across a broad gradient of stressor intensities.

Our study explores various ways in which different stressor intensities and stressor-effect relationships could combine

to cause mortality in a single population. However, we recognize that the realized response of populations to multiple stressors will also depend on the ecological context, including how the stressors impact biotic interactions and the population dynamics of the focal species. For example, Hodgson *et al*. [46] demonstrated that the form of density dependence operating on a population impacts the severity of multiple stressors in a given population. Furthermore, stressors can impact interspecific relationships, including resource competition [47], predator–prey relationships [48] or pathogen/parasite–host relationships [49], which may have larger impacts than the direct effects of the stressor on the focal population. While an increased focus on ecological complexity is clearly required to improve the utility of multiple stressor research for ecosystem management [23], producing predictive models for natural systems would be supported by a stronger understanding of the way in which multiple stressors act on single populations.

Data accessibility. The code used to produce and analyse the simulation component of this paper, including the code used to produce the figures in our manuscript, is available at https://github.com/cjdey/null_models_for_multiple_stressors. The code used to perform the meta-analysis is also available at the same link. The data used to perform the meta-analysis was kindly provided by Ben Halpern [19] and by Emily Darling [14], and can be requested by contacting them.

Authors' contributions. Developed the study idea: C.J.D. with assistance from M.A.K. Collected the data and conducted the analysis: C.J.D. Wrote the manuscript: C.J.D. with assistance from M.A.K. Revised the manuscript: C.J.D. and M.A.K.

Competing interests. The authors declare no competing interests.

Funding. This study was funded by Fisheries and Oceans Canada's Freshwater Habitat Science Initiative.

Acknowledgements. We thank Ben Halpern and Emily Darling for sharing data from their earlier meta-analyses. We also thank Emma Hodgson, Andrew Drake, Rowshyra Castañeda, Cindy Chu and Adam van der Lee for comments on earlier versions of this manuscript. Sasha Dall, Ralf Schäfer and one anonymous reviewer provided helpful reviews.

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
