## [Peer Review File · Proceedings of the Royal Society B: Biological Sciences]

Review History

RSPB-2020-3126.R0 (Original submission)

Review form: Reviewer 1 (Ralf Schaefer)

Recommendation

Accept with minor revision (please list in comments)

Scientific importance: Is the manuscript an original and important contribution to its field?

Excellent

General interest: Is the paper of sufficient general interest?

Excellent

Quality of the paper: Is the overall quality of the paper suitable?

Excellent

Is the length of the paper justified?

Yes

Should the paper be seen by a specialist statistical reviewer?

No

Do you have any concerns about statistical analyses in this paper? If so, please specify them explicitly in your report.

No

It is a condition of publication that authors make their supporting data, code and materials available - either as supplementary material or hosted in an external repository. Please rate, if applicable, the supporting data on the following criteria.

Is it accessible?

No

Is it clear?

Yes

Is it adequate?

Yes

Do you have any ethical concerns with this paper?

No

Comments to the Author

The authors analyse the predictive power of different null models for joint stressors as well as model and compare their predictions for simulated stressor intensities and shapes. The paper is very well written and very original including a comprehensive investigation of the subject matter. It should be of high interest to a wide range of multiple stressor researchers and will certainly make an impact. The authors are to be congratulated for this relevant contribution to the literature and for providing the computer code (though I could not run it as I have trouble with my tidyverse package)

78 „understanding which effect is expected is critical“ - please fix grammar

Figure 4 Could you provide information on how good each model predicted the joint effects within a range (e.g. $\pm 5\%$ or so). This information could be useful.

Ralf Schäfer

Review form: Reviewer 2

Recommendation

Accept with minor revision (please list in comments)

Scientific importance: Is the manuscript an original and important contribution to its field?

Excellent

General interest: Is the paper of sufficient general interest?

Good

Quality of the paper: Is the overall quality of the paper suitable?

Excellent

Is the length of the paper justified?

Yes

Should the paper be seen by a specialist statistical reviewer?

No

Do you have any concerns about statistical analyses in this paper? If so, please specify them explicitly in your report.

No

It is a condition of publication that authors make their supporting data, code and materials available - either as supplementary material or hosted in an external repository. Please rate, if applicable, the supporting data on the following criteria.

Is it accessible?

N/A

Is it clear?

N/A

Is it adequate?

N/A

Do you have any ethical concerns with this paper?

No

Comments to the Author

This paper uses simulations and existing data from published meta-analysis to determine how null model selection changes predictions about multiple stressor effects. The authors conclude that the different null models make different predictions, which is not surprising since that is the purpose of having different models! However, the argument that the combined effects of stressors should not be considered 'surprising' (i.e. antagonistic / synergistic) if the observed effect falls within the range of null model predictions is very compelling. Overall, I think the paper is very timely and nicely written - I only have a few minor suggestions:

1. Line 72: relative to the sum of the independent effects of each stressor?
2. Figure 1: This is really great!
3. Line 148: it is not very clear how the data was simulated, how realistic it is, and what it is meant to represent? Please can you add some more details on this?
4. Precautionary models - how these are defined needs to be clearer
5. Line 259: I don't think it is an assumption of these models that there isn't a mechanistic interaction between stressors? For instance, a synergistic interaction could be caused by warming causing a pollutant to be taken up faster. I see what you are saying on 306-308, but there are some scenarios where a response can fall within the range predicted by one of the 5 models because there is a mechanistic interaction between the stressors.
6. Line 263: I am not sure you do demonstrate this, especially as the data is simulated. Is this the purpose of the paper? Seems a bit out of place to state this here.

Decision letter (RSPB-2020-3126.R0)

15-Jan-2021

Dear Dr Dey:

Your manuscript has now been peer reviewed and the reviews have been assessed by an Associate Editor. The reviewers' comments (not including confidential comments to the Editor)

and the comments from the Associate Editor are included at the end of this email for your reference. As you will see, the reviewers and the Editors have raised some concerns with your manuscript and we would like to invite you to revise your manuscript to address them.

Research ethics:

Use of animals and field studies:

It is a condition of publication that you make available the data and research materials supporting the results in the article. Please see our Data Sharing Policies (<https://royalsociety.org/journals/authors/author-guidelines/#data>). Datasets should be deposited in an appropriate publicly available repository and details of the associated accession number, link or DOI to the datasets must be included in the Data Accessibility section of the article (<https://royalsociety.org/journals/ethics-policies/data-sharing-mining/>). Reference(s) to datasets should also be included in the reference list of the article with DOIs (where available).

If you wish to submit your data to Dryad (<http://datadryad.org/>) and have not already done so you can submit your data via this link [http://datadryad.org/submit?journalID=RSPB&manu=\(Document not available\)](http://datadryad.org/submit?journalID=RSPB&manu=(Document%20not%20available)), which will take you to your unique entry in the Dryad repository.

Please submit a copy of your revised paper within three weeks. If we do not hear from you within this time your manuscript will be rejected. If you are unable to meet this deadline please let us know as soon as possible, as we may be able to grant a short extension.

Best wishes,
Dr Sasha Dall
<mailto:proceedingsb@royalsociety.org>

Associate Editor
Board Member: 1
Comments to Author:
Dear authors,

Two reviewers and myself have read the MS. We all think it is potentially an important contribution that could fit well within the scope of the journal, as it tackles a very general question. The combination of simulations and re-analyzing a empirical large dataset is also appealing. Both reviewers only have relatively minor, but useful comments. However, I do have quite some points, as I felt the MS can still be improved in quite some ways and be made more accessible to a broader audience. Therefore I recommend a 'revision'.

Associate editor comments:

Major comments:

1. The title and introduction sets up the scope of this MS quite broad, but all the results focus on mortality. I can see that this study could be relevant beyond mortality, but at the moment this is not made very specific in the Discussion. Also I wondered whether in its current form the title is too broad and should include reference to mortality studies.
2. Throughout the MS I struggled at points how to relate it to my own research and field, which is in nonexperimental studies on wildlife population ecology, in which I also work on cumulative effects and survival analysis. This may be due to a difference in terminology or approaches used, and possibly my misreading at sections. However, the MS raises a very relevant point about null model selection, and it should be made more easily accessible to a wider audience of readers, such as myself.

3. Discussion of the additivity model may be too narrow. In my experience studies on mortality or survival that are in the realm of population and conservation ecologists typically do not test for additivity as

$L(bA \cdot SIA + bB \cdot SIB) = L(bA \cdot SIA) + L(bB \cdot SIB)$, what they do is they compare the following two models:

(a) $L(\text{Survival}) = bA \cdot SIA + bB \cdot SIB$

(b) $L(\text{Survival}) = bA \cdot SIA + bB \cdot SIB + bAB \cdot SIA \cdot SIB$

Where L is some link function that constrains survival or mortality to be between 0-1, e.g. logit in logistic regression (or probit, or..). Or in hazard models one would compare

$h(t) = h_0(t) \cdot \exp(bA \cdot SIA + bB \cdot SIB)$

with

$h(t) = h_0(t) \cdot \exp(bA \cdot SIA + bB \cdot SIB + bAB \cdot SIA \cdot SIB)$

This alternative way of thinking about additivity needs to be discussed, as it is a very common approach that is now missing from the scope of the MS. Maybe this difference in approach is related to experimental vs. observational studies, as in experimental studies one can manipulate SIA, SIB separately and SIA and SIB at the same time and look at patterns of mortality but in most ecological wildlife studies one would use spatiotemporal variation in SIA and SIB and use regression models as the ones described above to assess additivity.

An additional complexity is that additivity on the scale of the link function is something slightly different than additivity on original scale.

Minor comments:

1. L78 Break paragraph in two here, as this is a key point that is otherwise a bit buried in the middle of a paragraph?
2. L97 "it has assumptions that are unlikely to hold in most contexts(see e.g. Thompson et al. 2018)." The next sentence gives one very specific example for studies on mortality, but it remains unclear to the reader why it is unlikely to hold in most contexts and what other implausible assumptions are made, without having to read the Thompson paper. As this is such an important point for the paper it would be good to elaborate and be more specific.
3. L105: is this not making a bit of a caricature of the literature, as most survival analysis (logistic analysis, hazard model) use some link function that constrains mortality values to be between 0-1(00%) and additivity is assumed to act on the scale at which the regression function is analyzed? Related to this: L165 Could one argue that any approach that test for joint effects that can predict outcomes outside the possible range (ie. >100%) is not the most useful null model to start with?
4. L155: stress capacity->intensity? And it would be useful to see how such a distribution looks like, for example by adding it as a histogram in the background of the example figures in Box 1
5. L213 "as expected", this can of course be easily seen to be always true by comparing the equations in table mathematically. This could be done mathematically for more pair-wise comparisons of methods (e.g. in a supplement) to provide general insights. Possibly this may also show why L269 "we show that the Dominance model always produces the lowest mortality predictions."
6. Fig. 2b is not the most insightful, we see a lot of datapoints filling up parts of the parameter space, but for example in the first panel of 2b we cannot see what type of sims led to positive and what to negative values. The same can be said of Table 2, the pseudo-R2 give some idea of how they depend on the data structure, but mechanistically we gain little understanding of why this is so.
7. In the Introduction some more explanation may be needed why most other models are compared to the simple addition model, ie. Why this was taken as the reference (eg. In Fig. 2b and the meta-analysis)
8. L276: Nonetheless, we also found that the intensity of individual stressors remains the strongest predictor of the magnitude of the joint effect across all null models (Table 2). Is this correct, as table does not look at the average joint effect across all null models, but at its variability.
9. For the meta-analysis in Fig. 4, would it be insightful to also mention how many datasets provided evidence for joint stressors when using either of the 5 methods?

10. L293: this recommendation assumes that all included null-models are relevant in some way ecologically, as when one would include silly null-models these may also widen the range, but then taking the range across all null models as advised here may not be advisable and very conservative. This recommendation may thus depend on the relevance of the null models considered, but how can we assess this?

11. L308-311: I was really struggling to understand the reasoning behind this argumentation. If we know that mechanistically there is an interaction among stressors, and the phenomenological approach does not pick this up, does this then not suggest that the phenomenological approach is problematic (or contains irrelevant null models)? I may be missing something here or misread it due to my background in non-experimental wildlife ecology, but would be good to clarify the reasoning and/or text a bit more.

Reviewer(s)' Comments to Author:

Referee: 1

Comments to the Author(s)

The authors analyse the predictive power of different null models for joint stressors as well as model and compare their predictions for simulated stressor intensities and shapes. The paper is very well written and very original including a comprehensive investigation of the subject matter. It should be of high interest to a wide range of multiple stressor researchers and will certainly make an impact. The authors are to be congratulated for this relevant contribution to the literature and for providing the computer code (though I could not run it as I have trouble with my tidyverse package)

78 „understanding which effect is expected is critical“ - please fix grammar

Figure 4 Could you provide information on how good each model predicted the joint effects within a range (e.g. $\pm 5\%$ or so). This information could be useful.

Ralf Schäfer

Referee: 2

Comments to the Author(s)

This paper uses simulations and existing data from published meta-analysis to determine how null model selection changes predictions about multiple stressor effects. The authors conclude that the different null models make different predictions, which is not surprising since that is the purpose of having different models! However, the argument that the combined effects of stressors should not be considered 'surprising' (i.e. antagonistic / synergistic) if the observed effect falls within the range of null model predictions is very compelling. Overall, I think the paper is very timely and nicely written - I only have a few minor suggestions:

1. Line 72: relative to the sum of the independent effects of each stressor?
2. Figure 1: This is really great!
3. Line 148: it is not very clear how the data was simulated, how realistic it is, and what it is meant to represent? Please can you add some more details on this?
4. Precautionary models - how these are defined needs to be clearer
5. Line 259: I don't think it is an assumption of these models that there isn't a mechanistic interaction between stressors? For instance, a synergistic interaction could be caused by warming causing a pollutant to be taken up faster. I see what you are saying on 306-308, but there are some scenarios where a response can fall within the range predicted by one of the 5 models because there is a mechanistic interaction between the stressors.
6. Line 263: I am not sure you do demonstrate this, especially as the data is simulated. Is this the purpose of the paper? Seems a bit out of place to state this here.

Author's Response to Decision Letter for (RSPB-2020-3126.R0)

See Appendix A.

Decision letter (RSPB-2020-3126.R1)

19-Feb-2021

Dear Dr Dey

I am pleased to inform you that your manuscript RSPB-2020-3126.R1 entitled "The consequences of null model selection for predicting the joint effects of multiple stressors" has been accepted for publication in Proceedings B.

The referee(s) have recommended publication, but also suggest some minor revisions to your manuscript. Therefore, I invite you to respond to the referee(s)' comments and revise your manuscript. Because the schedule for publication is very tight, it is a condition of publication that you submit the revised version of your manuscript within 7 days. If you do not think you will be able to meet this date please let us know.

Online supplementary material will also carry the title and description provided during submission, so please ensure these are accurate and informative. Note that the Royal Society will

not edit or typeset supplementary material and it will be hosted as provided. Please ensure that the supplementary material includes the paper details (authors, title, journal name, article DOI). Your article DOI will be 10.1098/rspb.[paper ID in form xxxx.xxxx e.g. 10.1098/rspb.2016.0049].

Sincerely,
Dr Sasha Dall
Editor, Proceedings B
<mailto:proceedingsb@royalsociety.org>

Associate Editor:
Board Member
Comments to Author:

Dear authors,

Thank you for the revision, which I think has satisfactorily dealt with the minor comments of the reviewers and my own comments. I think the MS is nearly ready for publication, with only a few minor points remaining (see below).

Minor points:

1. Box 1 at the end now explains that this manuscript focusses on studies using the predictive null model approach and not on studies that use a regression approach. However, this distinction is not specifically referred to in the main text. This is likely to leave some readers wondering how this study relates to their own work if they are more familiar with observational studies that take

statistical regression approaches to look at joint effects. It would be good to add a sentence on this specifically early in the Introduction (with reference to Box 1), to clarify this as early as possible in the MS, as this will help to set the scope of the MS.

2. L288 that-> than

3. L289 Ddominance

4. L302-307: it would still be useful to explain (mechanistically) how such joint effects then do come about, they do not appear out of thin air.

5. L396: very long sentence

Author's Response to Decision Letter for (RSPB-2020-3126.R1)

See Appendix B.

Decision letter (RSPB-2020-3126.R2)

24-Feb-2021

Dear Dr Dey

I am pleased to inform you that your manuscript entitled "The consequences of null model selection for predicting the joint effects of multiple stressors" has been accepted for publication in Proceedings B.

Open Access

Paper charges

Sincerely,
Editor, Proceedings B
<mailto:proceedingsb@royalsociety.org>

Appendix A

Associate Editor
Board Member: 1
Comments to Author:
Dear authors,

Two reviewers and myself have read the MS. We all think it is potentially an important contribution that could fit well within the scope of the journal, as it tackles a very general question. The combination of simulations and re-analyzing a empirical large dataset is also appealing. Both reveiwers only have relatively minor, but useful comments. However, I do have quite some points, as I felt the MS can still be improved in quite some ways and be made more accessible to a broader audience. Therefore I recommend a 'revision'.

Associate editor comments:
Major comments:

1. The title and introduction sets up the scope of this MS quite broad, but all the results focus on mortality. I can see that this study could be relevant beyond mortality, but at the moment this is not made very specific in the Discussion. Also I wondered whether in its current form the title is too broad and should include reference to mortality studies.

Thank you for the suggestion. We have altered the title of our paper to clarify that we focus on the prediction of mortality resulting from multiple stressors, and have clarified several places in the text to make it more clear that we focused on mortality in our study. However, as the editor mentions, our results do provide some general insight beyond mortality. In fact, 4 of the 5 null models we explore have direct application to continuous endpoints such as growth or reproduction. Only the Stressor Addition Model does not have a direct application to continuous endpoints, because it considers mortality a consequence of exceeding an individual's general stress capacity. We now mention (on lines 136-140) that our simulation results should be generalizable to other endpoint types, with the exception of any inference from the Stressor Addition Model.

2. Throughout the MS I struggled at points how to relate it to my own research and field, which is in nonexperimental studies on wildlife population ecology, in which I also work on cumulative effects and survival analysis. This may be due to a difference in terminology or approaches used, and possibly my misreading at sections. However, the MS raises a very relevant point about null model selection, and it should be made more easily accessible to a wider audience of readers, such as myself.

Thank you for your support regarding our main results, and for highlighting your concern related to the accessibility of this paper. Indeed, in their recent review of the multiple stressor literature, Orr et al. (2020. *Proc B* 287) highlighted the significant divides between different fields as a major force hindering progress on understanding multiple stressors/cumulative effects.

We have endeavored to improve the accessibility of our paper by (i) clearly outlining the relationship between a predictive null model approach (as used in our paper) and a regression based approach (now in Box 1), (ii) expanding our meta-analysis to explore the accuracy of different null models in predicting joint stressor effects, (iii) improving the description of our simulation approach, (iv) moving a complex figure (former figure 2B) to the supplemental information, and (v) improving clarity and streamlining the discussion. We hope these measures, in addition to the glossary of terms provided in

Box 1, will help to make this material accessible to a wide audience despite its technical nature.

3. Discussion of the additivity model may be too narrow. In my experience studies on mortality or survival that are in the realm of population and conservation ecologists typically do not test for additivity as $f_{AB}(SIA, SIB) = f_A(SIA) + f_B(SIB)$, what they do is they compare the following two models:

(a) $L(\text{Survival}) = b_A * SIA + b_B * SIB$

(b) $L(\text{Survival}) = b_A * SIA + b_B * SIB + b_{AB} * SIA * SIB$

Where L is some link function that constrains survival or mortality to be between 0-1, e.g. logit in logistic regression (or probit, or..). Or in hazard models one would compare

$h(t) = h_0(t) * \exp(b_A * SIA + b_B * SIB)$

with

$h(t) = h_0(t) * \exp(b_A * SIA + b_B * SIB + b_{AB} * SIA * SIB)$

This alternative way of thinking about additivity needs to be discussed, as it is a very common approach that is now missing from the scope of the MS. Maybe this difference in approach is related to experimental vs. observational studies, as in experimental studies one can manipulate SIA, SIB separately and SIA and SIB at the same time and look at patterns of mortality but in most ecological wildlife studies one would use spatiotemporal variation in SIA and SIB and use regression models as the ones described above to assess additivity.

An additional complexity is that additivity on the scale of the link function is something slightly different than additivity on original scale.

Thank you for your comment. Indeed, as the Editor mentions, we have not consider all possible null models in our study, and have instead focused on the 5 models suggested as mostly applicable to the *prediction* of multiple stressor effects by Schafer and Piggott (2018 *Glob Chang Biol*). Other null models may be suitable depending on the type of data collected and the goals of the analysis (e.g. hypothesis testing or prediction of joint effects). We now make this point on lines 128-132.

Importantly, these models can all be used to make predictions based on the observed stress effect of the individual stressors, and do not require actual measurement of the stressor intensities. Indeed, this is the approach we use in the re-analysis of two meta-datasets where we only had data on the observed stress effects. The regression framework outlined by the Editor is different in that it focuses on the stressor intensities and does not require the individual stressor effects. Furthermore, the inclusion of link functions (or data transformations) alter the assumptions of the models. For example, conducting an ANOVA (without an interaction effect) on log-transformed values imposes the Multiplicative null model (see Griffen et al. 2016 *Marine Ecology Progress Series*, 543) rather than the Simple Addition model, a distinction that is often ignored. Finally, an unstated assumption of the regression framework is that the individual stressor-effect relationships are linear (or linear on the link scale) with a slope equivalent to the regression coefficient, however different shapes in the stressor-effect relationship are possible and warrant consideration. We now include a discussion of the difference between a regression approach and the predictive null model approach we used, in Box 1.

The Editor also raises another important point – the term ‘additivity’ has different meanings to different people, and a variety of different null models are called ‘additive’ (despite making very different predictions). For this reason, we tried to avoid the term ‘additivity’ in our paper, and instead refer to our reference model as the ‘Simple Addition

model'. (In our revised manuscript we removed a few instances of the term 'additivity' that were outstanding). The examples raised by the Editor (of additivity on the link scale, and additivity in the hazard model), are different than the Simple Addition model in our study, but consistent with our general message that there are a variety of possible null models and a multi-model approach would be beneficial.

Minor comments:

1. L78 Break paragraph in two here, as this is a key point that is otherwise a bit buried in the middle of a paragraph?

Done

2. L97 "it has assumptions that are unlikely to hold in most contexts(see e.g. Thompson et al. 2018)." The next sentence gives one very specific example for studies on mortality, but it remains unclear to the reader why it is unlikely to hold in most contexts and what other implausible assumptions are made, without having to read the Thompson paper. As this is such an important point for the paper it would be good to elaborate and be more specific.

Thank you for the suggestion. We have now modified this text to further explain the issues related to the simple addition model.

3. L105: is this not making a bit of a caricature of the literature, as most survival analysis (logistic analysis, hazard model) use some link function that constrains mortality values to be between 0-1(00%) and additivity is assumed to act on the scale at which the regression function is analyzed? Related to this: L165 Could one argue that any approach that test for joint effects that can predict outcomes outside the possible range (ie. >100%) is not the most useful null model to start with?

This difference is related to the different approaches used in experimental and non-experimental studies as identified by the Editor earlier in his comments, as well as to differences in the approach taken by researchers vs natural resource managers. Typically, experimental studies are not analyzed using a link function and the various meta-analyses conducted on experimental studies also do not use link functions. While link functions (and data transformations) improve the way that data interact with our statistical tests, they make predictions (and therefore, decision-making by resource managers) more challenging. For example, conducting an ANOVA (without an interaction effect) on log-transformed values imposes the Multiplicative null model (see Griffen et al. 2016 *Marine Ecology Progress Series*, 543), but the relationship between link functions / data transformations and the actual null model being tested is not always clearly understood.

We agree that any model that predicts outcomes outside of the possible range (i.e. >100%) is not the most useful null model. We only discuss it in the context of the Simple Addition model in this paper, because that is the only one of the five null models we considered which has this feature.

4. L155: stress capacity->intensity? And it would be useful to see how such a dsitrubution looks like, for example by adding it as a histogram in the background of the example figures in Box 1

The sentence correctly refers to stress capacities (F_{strcap}), which are a component of the stressor addition model (see Table 1). As suggested, we now include a figure illustrating this distribution. It is included as Figure S2 in the supplementary information.

5. L213 “as expected”, this can of course be easily seen to be always true by comparing the equations in table mathematically. This could be done mathematically for more pair-wise comparisons of methods (e.g. in a supplement) to provide general insights. Possibly this may also show why L269 “we show that the Dominance model always produces the lowest mortality predictions.”

As the Editor mentions, analytical comparison of the equation for the Simple Addition with the Dominance and Multiplicative models is possible, and we now mention this insight on line 178-181. However, we do not believe that an such approach is possible for the Concentration Addition and Stressor Addition models, given the complexities of (i) the various stressor-effect relationships considered in the concentration addition model, and (ii) the need to consider the cumulative density function of the stress capacities for the stressor addition model. For these reasons, we selected a simulation approach that is applicable to all model comparisons. We now outline this reasoning in our manuscript on lines 182-184

6. Fig. 2b is not the most insightful, we see a lot of datapoints filling up parts of the parameter space, but for example in the first panel of 2b we cannot see what type of sims led to positive and what to negative values. The same can be said of Table 2, the pseudo-R2 give some idea of how they depend on the data structure, but mechanistically we gain little understanding of why this is so.

Thank you for the suggestion. We have now moved Figure 2B to the supplemental information section. We would prefer to retain Table 2 in the main text, and now provide more explanation that these results indicate the strength of each of the variables in predicting the amount of mortality, across all 5 null models.

7. In the Introduction some more explanation may be needed why most other models are compared to the simple addition model, ie. Why this was taken as the reference (eg. In Fig. 2b and the meta-analysis)

We have added some further explanation to the lines 144-147. In brief, we made this decision because Simple Addition is the most common null model used in experimental studies of multiple stressors and is therefore the best reference to provide insight on the consequences of selecting alternative null models.

8. L276: Nonetheless, we also found that the intensity of individual stressors remains the strongest predictor of the magnitude of the joint effect across all null models (Table 2). Is this correct, as table 2 does not look at the average joint effect across all null models, but at its variability.

We have now clarified that the pseudo R^2 values listed are measures of the strength of each variable in driving the amount of predicted mortality (across all null models). As such, higher values indicate that the variable is a stronger predictor of the model predicted mortality. See lines 254-257 and the caption of Table 2.

9. For the meta-analysis in Fig. 4, would it be insightful to also mention how many datasets provided evidence for joint stressors when using either of the 5 methods?

The Editor's comment is a bit unclear to us, as we are not sure what is meant by 'evidence for joint stressors'. However, in response to a comment on this analysis from reviewer 1, we have now added additional information to this section of the paper. Specifically, we now include an analysis of the accuracy (bias and prediction) of each of the null models in predicting the observed joint stressor effects from the meta-analyses, which explores how well each of the null models do in predicting the observed joint effects. This analysis is outlined on lines 220-225 and in Figure 4B.

10. L293: this recommendation assumes that all included null-models are relevant in some way ecologically, as when one would include silly null-models these may also widen the range, but then taking the range across all null models as advised here may not be advisable and very conservative. This recommendation may thus depend on the relevance of the null models considered, but how can we assess this?

The Editor raises a good point. We suggest that the 5 null models used here are all ecologically relevant, given that they have been previously applied to the study of multiple stressors in ecology or ecotoxicology (see also Schafer and Piggott 2018 for further discussion of the application of these models to multiple stressors). As in other multi-model analyses in ecology, theoretical considerations can be used to select a reasonable set of models for further analysis. We suggest an analogous approach could be used to create an interval of null model predictions for multiple stressors. We have modified the text to better explain this idea. See lines 343-351.

11. L308-311: I was really struggling to understand the reasoning behind this argumentation. If we know that mechanistically there is an interaction among stressors, and the phenomenological approach does not pick this up, does this then not suggest that the phenomenological approach is problematic (or contains irrelevant null models)? I may be missing something here or misread it due to my background in non-experimental wildlife ecology, but would be good to clarify the reasoning and/or text a bit more.

The phenomenological approach to the prediction of multiple stressor effects is not interested in whether stressors have a mechanistic interaction or act independently. Instead, it makes predictions using previously observed patterns and in the absence of information about mechanism. As such, a phenomenological approach can only be problematic if it does a poor job in predicting the outcome, and specifically, if it does a poorer job than an approach that is informed by details of the mechanism of stressor interactions. Our argument is that our current understanding of the mechanistic interaction between stressors is not sufficiently advanced to provide robust prediction of the effects of multiple stressors. As such, a phenomenological approach is the current best method (although this will hopefully change as more information on mechanisms is discovered).

We have revised the paragraph in question to better explain our argument.

Reviewer(s)' Comments to Author:

Referee: 1

Comments to the Author(s)

The authors analyse the predictive power of different null models for joint stressors as well as model and compare their predictions for simulated stressor intensities and shapes. The paper is very well written and very original including a comprehensive investigation of the subject matter. It should be of high interest to a wide range of multiple stressor researchers and will certainly make an impact. The authors are to be congratulated for this relevant contribution to the literature and for providing the computer code (though I could not run it as I have trouble with my tidyverse package)

We thank Dr. Schäfer for his comments and encouragement. Indeed, this paper was strongly motivated by his work with Dr. Piggott and we are pleased that he believes it will be a valuable contribution.

78 „understanding which effect is expected is critical“ - please fix grammar

Done

Figure 4 Could you provide information on how good each model predicted the joint effects within a range (e.g. $\pm 5\%$ or so). This information could be useful.

Thank you for the suggestion. We have now added additional information to this section of the paper. Specifically, we include an analysis of the accuracy (bias and prediction) of each of the null models in predicting the observed joint stressor effects, which explores how well each of the null models predict the observed joint effects. See the new panel included in figure 4.

Ralf Schäfer

Referee: 2

Comments to the Author(s)

This paper uses simulations and existing data from published meta-analysis to determine how null model selection changes predictions about multiple stressor effects. The authors conclude that the different null models make different predictions, which is not surprising since that is the purpose of having different models! However, the argument that the combined effects of stressors should not be considered ‘surprising’ (i.e. antagonistic / synergistic) if the observed effect falls within the range of null model predictions is very compelling. Overall, I think the paper is very timely and nicely written - I only have a few minor suggestions:

1. Line 72: relative to the sum of the independent effects of each stressor?

Good suggestion, we have changed the wording to reflect the reviewer’s suggestion.

2. Figure 1: This is really great!

Thank you

3. Line 148: it is not very clear how the data was simulated, how realistic it is, and what it is meant to represent? Please can you add some more details on this?

We now provide more information related to our simulations and clarify some sections of text that may have been confusing. See lines 164-176. We also note that we have included the data analysis script so that interested readers can reproduce the simulations at their leisure.

4. Precautionary models – how these are defined needs to be clearer

We now provide more explanation on lines 147-151 and lines 269-274

5. Line 259: I don't think it is an assumption of these models that there isn't a mechanistic interaction between stressors? For instance, a synergistic interaction could be caused by warming causing a pollutant to be taken up faster. I see what you are saying on 306-308, but there are some scenarios where a response can fall within the range predicted by one of the 5 models because there is a mechanistic interaction between the stressors.

The suite of null models considered in this study are all built upon the independent (i.e. non-interacting) action of two stressors. In this definition, we follow the work of Schafer and Piggott (2018, *Glob Chang Biol*) outline the general motivation and reasoning for applying null models to the study of multiple stressors :

“Stressor interactions are determined in reference to a null model that predicts the joint effect assuming the absence of interactions, that is, the stressors are operating independently.” (Schafer and Piggot, 2018 p 1818).

As the reviewer mentions, it is possible for stressors to have a mechanistic interaction and yet still produce a joint effect that falls within the range of the predictions of the null models. We mention this possibility on lines 308-310 of the original submission (now lines 359-361).

6. Line 263: I am not sure you do demonstrate this, especially as the data is simulated. Is this the purpose of the paper? Seems a bit out of place to state this here.

As stated above, the null models presented in this paper do not include a mechanistic interaction between stressors. Since we demonstrate that different null models produce a range of joint effects, we believe that this is evidence to support our statement ‘that direct interactions between stressors are not required to produce a range of joint effects’. We have revised the sentence to indicate that this is a theoretical consideration, since (as the reviewer states), we do not have empirical data on mechanistic interactions in which to test this idea.

Appendix B

1. Box 1 at the end now explains that this manuscript focusses on studies using the predictive null model approach and not on studies that use a regression approach. However, this distinction is not specifically referred to in the main text. This is likely to leave some readers wondering how this study relates to their own work if they are more familiar with observational studies that take statistical regression approaches to look at joint effects. It would be good to add a sentence on this specifically early in the Introduction (with reference to Box 1), to clarify this as early as possible in the MS, as this will help to set the scope of the MS.

Thank you for the suggestion. We now add mention that among-study differences in analytical approach present a further complication for the consideration of null models, and refer readers to Box 1. See lines 92-95.

2. L288 that-> than

Done

3. L289 Ddominance

Done

4. L302-307: it would still be useful to explain (mechanistically) how such joint effects then do come about, they do not appear out of thin air.

Thank you for the suggestion. We now refer the readers to the seminal paper by Schäfer and Piggott 2018 who outline the details of these null models and explain how they can produce different joint effects without mechanistic interaction between the stressors themselves. We provide some explanation earlier in the manuscript (Lines 116-131) which also explains how different null models produce different joint effects without mechanistic interactions between the stressors.

5. L396: very long sentence

This sentence has now been broken in two.